# Study on a Prediction Model of Superhighway Fuel Consumption Based on the Test of Easy Car Platform

**Yong-Ming He [1,2,*], Jia Kang [1,3,*], Yu-Long Pei [1,3], Bin Ran [2,4] and Yu-Ting Song [1,3]**

1 School of Transportation, Northeast Forestry University, 26 Hexing Road, Harbin 150040, China; peiyulong@nefu.edu.cn (Y.-L.P.); songyuting@nefu.edu.cn (Y.-T.S.)

2 School of Civil Engineering, University of Wisconsin-Madison, Madison, WI 57305, USA; yhe275@wisc.edu

3 Transport Research Centre, Northeast Forestry University, 26 Hexing Road, Harbin 150040, China

4 Southeast University-University of Wisconsin Intelligent Network Transportation Joint Research Institute, 2312 Engineering Hall, 1415 Engineering Drive, Madison, WI 53706, USA

* Correspondence: hymjob@nefu.edu.cn (Y.-M.H.); kangjia@nefu.edu.cn (J.K.);
  Tel.: +86-136-3360-2189 (Y.-M.H.); +86-187-5519-5476 (J.K.)

**Abstract:** To explore the relationship between fuel consumption and speed for a vehicle on a superhighway with a design speed exceeding 120 km/h, the fuel consumption data provided by the Test of Easy Car platform are used to fit the fuel consumption of different models. The fitting results show that the fitting degree of fuel consumption by a cubic curve is the highest, and the correlation coefficient is above 0.95. A fuel consumption cubic curve model of different vehicle types is established by using the fitting parameters to predict the fuel consumption when a vehicle is running at a speed of 130 km/h–180 km/h. The prediction results show that the average fuel consumption of compact vehicles is the lowest when a vehicle is running on a superhighway at speeds of 130 km/h–180 km/h, with values of 8.95 L/100 km–16.26 L/100 km; the average fuel consumption of sport utility vehicles (SUVs) is the highest, with values of 12.65 L/100 km–22.70 L/100 km. The prediction results can be used to estimate the cost of using a superhighway and provide a basis for estimating the feasibility of superhighways.

**Keywords:** superhighway; constant-speed fuel consumption; curve fitting; prediction model; comparative analysis

## 1. Introduction

The 1951 edition of the *Highway Engineering Design Guidelines (Draft)* first stipulated a maximum design speed for China's highways of 120 km/h. After nearly 70 years of development, the standard has been revised 8 times. The latest *Highway Engineering Technical Standard (JTG B01-2014)* still stipulates a maximum highway speed of 120 km/h. With the development of society and the economy, highway construction technology and new automobile technology have made leaps forward, and it is possible to construct superhighways with a design speed over 120 km/h. Based on an analysis of highway construction technology and new automobile technology, our research team proposed the concept of a superhighway after sufficient demonstration, that is, a highway with a design speed exceeding 120 km/h [1]. In addition, after many discussions among domestic experts, the classification of superhighways is put forward, as shown in Table 1.

**Table 1.** Classification of superhighways.

| Superhighway Grade | Grade III | | | Grade II | | | Grade I | | |
|---|---|---|---|---|---|---|---|---|---|
| Design speed (km/h) | 180 | 160 | 140 | 160 | 140 | 120 | 140 | 120 | 100 |

Construction of the first superhighway in China, the Hang-Shao-Yong highway, began in 2018, and it is expected to open in 2022 before the opening of the Asian Games in Hangzhou. The completion of this superhighway will be a milestone in the development history of China's superhighways. However, the development of superhighways inevitably faces the problem of energy consumption. At present, there are few studies on the prediction of the fuel consumption of vehicles on a superhighway whose design speed exceeds 120 km/h. In this paper, fuel consumption data of small vehicles, compact vehicles, mid-size vehicles and SUV (sport utility vehicle) models provided by the Test of Easy Car platform are fitted with multiple models. The Test of Easy Car platform is an automobile test website in China, which provides professional tests on the power, economy and safety of vehicles sold on the market. The results are published on its official website for consumers' reference. The test data is public and accessible via the Internet. In this paper, fuel consumption test data of the platform is selected for research. The fitting results show that the fitting degree of a cubic curve is the highest in different models of fuel consumption, and the correlation coefficient is above 0.95. The fuel consumption of vehicles conforms to the rule of a cubic curve. Therefore, a cubic curve is used to predict the fuel consumption of different models at speeds of 130 km/h–180 km/h. This work provides a reference for the fuel consumption prediction of vehicles on superhighways whose speeds exceed 120 km/h.

On the one hand, the established fuel consumption prediction model can provide a reference for predicting the consumption of fuel of vehicles on the superhighway with a design speed of over 120 km/h, and estimate the cost of vehicles on the superhighway. In addition, by establishing the models of different vehicle types and comparing the differences of fuel consumption performance of different vehicle types, this study can provide economic reference for road users when they choose different vehicle types. Finally, the difference in such performance is also related to vehicle exhaust emissions, which can be used as a reference for studying such emissions in different vehicle models.

## 2. Overview of Consumption Research on Superhighways

### 2.1. Research on Superhighways

At present, research on superhighways in China and abroad is still in its infancy. Our research team published several papers from 2016 to 2020 to conduct in-depth studies on the feasibility, safety, economy and traffic capacity of superhighways. In 2016, our research team studied the different stages of the development of new automobile technology and highway construction technology, demonstrated the feasibility of developing a superhighway, and conducted an in-depth study on the necessity of developing a superhighway [1]. Subsequently, the team analyzed the driving characteristics of a superhighway in 2017, conducted an in-depth study on the capacity of the superhighway, and demonstrated the role of autonomous driving technology in improving the traffic capacity of the superhighway [2]. The following year, to evaluate the economical efficiency of the superhighway, our research team conducted a comparative study on the travel expenses of superhighways and compared the single cost of traveling among civil aviation, ordinary highways, railways and other travel modes. The results showed that the single cost of traveling by superhighway is lower than that of an economy class ticket for civil aviation and is equivalent to a second-class seat for a high-speed railway and has certain economic advantages [3,4]. Additionally, the construction cost and use cost of a superhighway once again were studied by us in 2019, and the results showed that the comprehensive cost of a superhighway was higher than that of an ordinary highway and lower than that of a railway, and the economic feasibility of superhighways was again demonstrated [5]. In 2020, to demonstrate the feasibility of superhighways from the perspective of safety, we studied a virtual track technology based on intelligent road buttons, which provided a strong guarantee for the safety of driving on a highway [6]. In the same year, to demonstrate the safety alignment of superhighways, our research team studied the design theory of plane alignment, vertical alignment and cross-section alignment of superhighways based on the linear design theory of ordinary superhighways and the linear design of

railways. Calculations and analysis showed that the safety of superhighways can be guaranteed by means of adopting plane curves with a large radius and relatively gentle longitudinal slope [7].

Our team's research on superhighways has aroused wide interest from domestic scholars. Zhao et al. established an obstacle identification model to analyze the curves of superhighways and studied the radius of the safety curve of superhighways by investigating the design of the curve of an existing highway and the problems [8]. Chen et al. analyzed the advantages, disadvantages and external challenges of superhighways by means of SWOT (situation analysis based on internal and external competitive environment and competitive conditions) analysis during their construction and operation management, provided suggestions for the development of a superhighway, and demonstrated the advantages of developing a superhighway [9]. Although some concepts of superhighways have been put forward in foreign countries, most of those studies have focused on the information superhighway, transportation of biochemical substances and other aspects, and a real superhighway has not yet been involved [10,11].

At present, the maximum speed limit of highways in more than 20 European countries has reached 130 km/h; in Italy it is 140 km/h, and in Texas it is 137 km/h (85 mile/h). There are 129,993 km of highway in Germany, approximately 2/3 of which have no speed limit [12].

For example, 15 years ago, the speed limit of French motorways was set at 130 km/h. Long-term safe operation has provided rich experience for China's construction of superhighways with design speed exceeding 120 km/h. Recently, due to the change of climate conditions in Europe, the French Citizens' Climate Conference proposed to reduce the speed limit of motorways. However, this is only a drop in the ocean and cannot solve the problem fundamentally. In addition, with the rapid development of highway construction technology and automobile technology, autonomous driving technology has become increasingly mature and gradually industrialized. From Level 0 to Level 5, human intervention is less and less, the degree of intelligence is higher and higher, and the safety is more and more guaranteed. With the improvement of the level of automatic driving, safety hazards such as delayed reaction and wrong judgment caused by human physiological factors are less and less, or even eliminated, which greatly reduces the threat of adverse environment (such as rain and snow) to driving safety. Besides, Level 4 self-driving cars have entered the market, which can effectively guarantee the operation safety of superhighways. In late 1999, France established LAVIA (Limiteur s'Adaptant à la VItesse Autorisée), an intelligent speed adaptation (ISA) project. The generic term intelligent speed adaptation (ISA) includes a wide range of different technologies designed to improve road safety by reducing traffic speeds and homogenizing traffic flows within the limits of speed limits. It is estimated that a car equipped with such a speed adaptation system would reduce its injury rate under current conditions by 6 to 16 percent during use. The paper also makes a detailed analysis of the causes of fatal traffic accidents and concludes that overspeed (exceeding the speed limit) is an important cause of fatal traffic accidents. However, the "variable speed limit" system can be used to consider that speeds in certain parts of the road network should not be below the published speed limit, for example, at sharp turns, pedestrian crossings or collision black spots. The "dynamic speed limit" system takes further account of changes in weather and traffic conditions. The use of these systems can guarantee traffic safety caused by speeding [13].

Studies have shown that, before 1997, the traffic death rate in France and the United States were fundamentally the same. Since 1997, the traffic death rate in France and the United States has decreased significantly. However, the traffic death rate in the United States is still much higher than that in France. Studies have speculated that Princess Diana died in Paris in 1997 as a result of a traffic accident caused by drunk drivers and speeding. Under pressure from public opinion, the French government has strengthened the formulation of traffic safety policies, such as making drunk driving illegal and implementing traffic light radar system. After 1997, the traffic death rate in France has been decreasing year by year. Although the United States and France both have similar vehicle engineering, road conditions and driver education, the United States did not adopt effective traffic policies after 1997 as France did, resulting in a much higher traffic death rate than France. This explains the importance of

policy and traffic execution to traffic safety, which can guarantee the traffic safety problems caused by speeding to some extent [14].

Therefore, from the perspective of highway construction technology, vehicle performance and foreign experience, conditions are ripe for the construction of superhighways with design speed exceeding 120 km/h in China.

According to the current situation of highway development in foreign countries and the related research on superhighways in China, research on superhighways with design speeds over 120 km/h has a deep foundation, and the development prospects are quite considerable. The development of superhighways will bring opportunities and challenges. At present, the most serious problem in developing superhighways is safety and energy consumption. In this paper, through a comparative analysis of fuel consumption data of different models, a cubic curve model is established to provide a basis for prediction of fuel consumption on superhighways.

## 2.2. Research on Automobile Fuel Consumption

### 2.2.1. Research on Domestic Automobile Fuel Consumption

At present, a large amount of research has been conducted on the energy consumption of vehicles on highways in China and abroad. For example, Feng et al. studied the fuel consumption of commercial vehicles on mountain highways based on VSP (vehicle specific power) modeling and obtained the influence of vehicle weight on vehicle speed, which has a greater impact on fuel consumption [15]. Jia et al. performed a comparative analysis of the evaluation index of fuel consumption of vehicles on highways, and the results showed that, when the driving speed is the same, the fuel consumption of vehicles on highways is lower than that of vehicles on ordinary roads [16]. Peng et al. conducted an in-depth study on a fuel consumption model of automobiles on highways and modified the parameters of a traditional fuel consumption model by using experimental data to make the newly established fuel consumption model more accurate [17]. Guo et al. established a multivariate linear regression fuel consumption prediction model by using the principle of multivariate statistical analysis, taking the unit fuel journey as the dependent variable and vehicle weight and horsepower as the independent variables by means of statistical analysis software and a method of curve fitting [18,19]. Zhang et al. put forward a method that can accurately evaluate the fuel consumption of vehicles by using the least squares method on the basis of fully considering many nonlinear factors affecting the fuel consumption, and the method can further reduce the impact of other factors on the model accuracy [20].

The performance of the engine has the greatest impact on the fuel consumption of the vehicle at high speed. Generally speaking, the larger the displacement of the engine, the better the performance. The greater the output power, the better the dynamic performance of the vehicle. However, if the engine displacement is large, it may consume more fuel, and the fuel consumption of the car at high speed will also increase, especially for vehicles with a driving speed close to 200 km/h, such as Ruicheng CC and 2019 Honda Civic. In 2018, Ruicheng CC realized the release test at the speed of 200 km/h in the high ring runway. The car is equipped with ACC (adaptive cruise control) adaptive cruise, PAB (passenger airbag)active brake and LDW (lane departure warning)function, which makes the overall automatic driving level of the car to L2 level, which can ensure the safety of the vehicle under the premise of providing comfortable driving. In addition, the 1.5 T turbocharged engine equipped with Ruicheng CC has a maximum power of 156 HP (115 kW)/5500 rpm and a maximum torque of 225 nm/2000–4000 rpm, which makes the car have enough power to drive at high speed. From the perspective of driverless technology and engine performance, the driving safety of vehicles with a speed of more than 120 km/h can be guaranteed, and there is enough power for them to drive at high speed. Therefore, it is meaningful to study the fuel consumption of vehicles over 120 km/h. In addition, with the vigorous promotion of new energy vehicles, such as pure electric vehicles and hybrid electric vehicles, they have the common characteristics of clean, efficient, energy-saving and environmental protection. The popularization of new energy vehicles can offset the excessive energy consumption

caused by ultra-high speed driving and can effectively solve the energy consumption problem caused by the development of superhighway.

Due to the differences in domestic and foreign traffic laws and regulations, driving habits of drivers, speed limits of vehicles, etc., many factors affecting vehicle driving are quite different, so research on the fuel consumption of driving vehicles in foreign countries is different from that in China. Many fruitful results have been achieved abroad in long-term research, and these research ideas and methods are worthy of our reference.

2.2.2. Research on Fuel Consumption of Foreign Automobiles

Westarp found that the parameters of many existing fuel consumption models were not applicable with a change in conditions in the research process, especially a change in speed [21]. Therefore, a new model was proposed to represent the functional correlation between fuel consumption and speed, and the model was verified by actual collected data. Zhang et al. conducted research on the fuel consumption of hybrid electric vehicles, and a modified fuel consumption model was established based on the working principle of the internal combustion engine [22]. Furthermore, the energy consumption of ground traffic flow under the hybrid model was further studied, and a quantitative analysis of fuel consumption was performed. Eugen et al. mainly studied the relationship between engine control and fuel consumption [23]. The results showed that using a nonlinear model to control the engine operation can improve the fuel consumption efficiency, and a simulation analysis of the results was carried out. Lijun Hao et al. [24] estimated the fuel economy of passenger cars in high-altitude areas through taxiing experiments. On the basis of this research, a reverse simulation method was used to build a model of internal combustion engines through the experimental data of engine torque and speed. The model was verified by simulations with actual data, and the simulation results showed that the error degree of the model is very small and that the model is feasible. Gunawan et al. thought deeply about the current problems of global warming and increasing greenhouse gases and concluded that automobile exhaust emissions account for 20% of greenhouse gas emissions [25]. This paper mainly studied the influence of driving strategy on the fuel consumption of automobiles and performed a quantitative analysis of vehicle dynamics, and a correlation between braking distance and fuel consumption was obtained. Shahin et al. predicted additional fuel consumption by using an artificial neural network model and obtained a correlation among vehicle weight, fuel consumption and engine displacement through quantitative analysis [26]. Maya et al. analyzed and studied energy policy. Through a comparison of fuel consumption data, the differences in fuel consumption and its characteristics were studied, fuel consumption control was carried out, and an actual fuel economy standard was established [27]. Collier, Sonya et al. fully studied and analyzed data of transport trucks to estimate the average stroke fuel consumption [28]. The relationship between vehicle models and fuel consumption was obtained by comparing different types of trucks.

Vehicle design, road geometry, road conditions, speed limit, driving mode and other important factors have a great impact on vehicle fuel consumption. For example, the design of vehicle shape and tires will make the vehicle subject to different degrees of air resistance and rolling resistance during driving, which will consume the energy generated by vehicle fuel [29,30]. When the road conditions are good, the work done by the car to overcome the road resistance will be reduced, which in turn will reduce the fuel consumption. Under the influence of engine performance, road conditions, load, wind direction, climate and usage, the vehicle will consume the least amount of fuel when driving within a certain speed range, which is the economic speed range. Generally speaking, the geometry of different roads will also have an impact on the fuel consumption of vehicle. When there are more uphill sections and turning sections, the vehicle must overcome the uphill resistance or brake to decelerate the turning to achieve the driving purpose, which will increase fuel consumption to a certain extent [31].

In addition, there are many methods to predict the fuel consumption of passenger vehicles, such as artificial neural network method, multiple linear regression, least square method and simulation modeling method, etc. [19,20]. However, different methods have different advantages and

disadvantages, which must be selected according to different vehicle types, different road conditions, and different driving modes, etc. Most importantly, the prediction results should be verified, and error analyzed, and necessary corrections should be made to ensure the accuracy of the prediction results.

In this paper, the fuel consumption data of different models are analyzed, curve fitting and modeling are carried out by SPSS software to predict the amount of fuel vehicles will consume at speeds over 120 km/h, and the fuel consumption curves of different models are compared.

## 3. Collection of Fuel Consumption Data

### 3.1. Test Method for Fuel Consumption in the Test of Easy Car Platform

The accuracy of a fuel consumption tester is 0.01 mL. During a test, the flow meter of the fuel consumption tester is connected in series to an oil supply pipeline for the vehicle engine under test. All the gasoline supplied to the engine for combustion will be measured by our fuel consumption tester, so the fuel flow measured by the fuel consumption tester is the actual fuel consumption of the engine. The serial method solves the problem of inaccurate measurement of fuel consumption caused by the jumping shot method. In addition, considering the physical characteristics of "heat expansion and cold contraction" of fuel, the calibration calculation formula given by the national standard is used to correct the final result of the fuel consumption test, and an accurate result was obtained for the fuel consumption test. By means of these tests and treatments, the error of the fuel consumption test results will be greatly reduced.

When a constant-speed fuel consumption test is conducted, the test vehicle is driven at a uniform speed on a flat road surface. Four groups of tests are conducted for each speed level, and each group is tested for the fuel consumption at a distance of 500 m. Finally, the average value of the four groups is taken as the final fuel consumption value of a speed level. In this way, the random error can be reduced, and the accuracy of test data can be guaranteed. The speed levels in the test are divided into 40 km/h, 50 km/h, 60 km/h ... 120 km/h. The above treatment method can guarantee the accuracy of the measured data and improve the accuracy of the following data analysis.

### 3.2. Classification of the Test Vehicle

Through the Test of Easy Car platform, fuel consumption data of 50 models have been obtained in total, which can be broadly divided into several models such as small vehicles, compact vehicles, mid-size vehicles and SUVs; small vehicles: car length less than 4.3 m, wheelbase less than 2.5 m; compact vehicles: car length less than 4.6 m, wheelbase less than 2.7 m; mid-size vehicles: car length less than 4.9 m, wheelbase less than 2.8 m. The fuel consumption data mainly refer to the fuel consumption per 100 km (L/100 km) when the vehicle is driven at speeds of 40, 50, 60 ... 120 km/h.

Data of 5 brands of small vehicles, compact vehicles, mid-size vehicles and SUV models were input into SPSS software for curve fitting and data analysis. Regression analysis is carried out by SPSS software, and the model with the largest correlation coefficient for the four vehicle types is determined by curve fitting. Then, the respective fuel consumption model equations are established according to an estimation table of model parameters. Small vehicles, compact vehicles, mid-size vehicles and SUV models are numbered in English letters. The models and corresponding serial numbers are shown in Table 2.

**Table 2.** Model and number grouping.

| Number | Vehicle Type | Number | Vehicle Type |
|---|---|---|---|
| A | Dongfeng yueda Kia xiu er 1.6 L GL automatic | Z | Changan Suzuki Tianyu SX4 1.6 L urban fashion |
| B | Changan Suzuki Swift 1.5LSuper flash version flash sharp | G | Brilliance China junjie FRV 1.3 L Manual comfort |
| C | Tianjin faw xiali N5 1.3 L Manual luxury model | H | Dongfeng Honda civic 1.8 L EXi automatic comfort version |
| D | Great Wall dazzle 1.3 L Manual CROSS version | I | Beijing hyundai i30 1.6 L Automatic comfort type |
| E | Changan Ford new fiesta 1.5 L Manual fashion model | J | Saic roewe 550 1.8 L DVVT S Automatic start |
| K | Buick regal 2.4 T | P | Nissan xiao passenger CVT 2WD |
| L | Mazda rui wing 2.5 L Supreme edition | Q | Nissan qijun 2.5 L XL luxury edition |
| M | New jing cheng SX 2.0 T AT | R | Great Wall harvard H3 skylight |
| N | Audi A6 2.0 T | S | Kia lion run 2.0 T Automatic 2WD GLS |
| O | Passat new area 1.8 T Automatic | T | Modern Tucson 2.0 T Automatic skylight |

## 4. Data Analysis and Curve Fitting by SPSS Software

### 4.1. Fuel Consumption Model for Small Vehicles

#### 4.1.1. Fuel Consumption Data Collection for Small Vehicles

Small vehicles generally refer to the A0 class car in model A, with low fuel consumption and a compact model. The wheelbase of this car is generally between 2.2 m and 2.3 m (some models are beyond this range), and the engine displacement is 1 L to 1.3 L. As its cost performance is better, this is the first choice of many car users, for example, Charley, swift, and so on. The fuel consumption data of 5 kinds of small vehicles were collected through the Test of Easy Car platform. The model numbers were A, B, C, D and E. The fuel consumption data are shown in Table 3.

**Table 3.** Constant-speed fuel consumption of small vehicles (L/100 km).

| Speed (km/h) | 40 | 50 | 60 | 70 | 80 | 90 | 100 | 110 | 120 |
|---|---|---|---|---|---|---|---|---|---|
| A | 5.37 | 4.91 | 5.11 | 5.52 | 6.14 | 6.61 | 7.24 | 8.06 | 9.11 |
| B | 5.24 | 4.68 | 4.39 | 4.95 | 5.21 | 5.77 | 6.30 | 7.09 | 7.78 |
| C | 5.80 | 4.98 | 4.89 | 5.32 | 5.47 | 5.68 | 6.39 | 7.05 | 8.10 |
| D | 4.69 | 3.77 | 4.01 | 4.30 | 4.83 | 5.49 | 5.96 | 6.88 | 7.59 |
| E | 5.89 | 5.27 | 4.91 | 5.30 | 5.66 | 6.24 | 6.81 | 7.50 | 8.51 |

#### 4.1.2. Fitting and Analysis of Fuel Consumption Data of Small Vehicles

The fuel consumption data of small vehicles shown in Table 3 were imported into SPSS for curve fitting and data analysis, and the curve and model with the highest fitting degree for each model could be obtained; here, model B is taken as an example.

The model parameters and fitting curves of model B for small vehicles are shown in Table 4 and Figure 1.

**Table 4.** Model summary and parameter estimation of vehicle B.

| Model Equations | Model Summary | | | | | Parameter Estimation | | | |
|---|---|---|---|---|---|---|---|---|---|
| | R-Squared | F | $df_1$ | $df_2$ | Sig. | Constant | $b_1$ | $b_2$ | $b_3$ |
| Linear equation | 0.778 | 24.574 | 1 | 7 | 0.002 | 2.775 | 0.037 | | |
| Logarithmic curve | 0.641 | 12.524 | 1 | 7 | 0.009 | −4.926 | 2.460 | | |
| Reciprocal curve | 0.488 | 6.666 | 1 | 7 | 0.036 | 7.734 | −143.291 | | |
| Conic | 0.974 | 110.213 | 2 | 6 | 0.000 | 7.428 | −0.093 | 0.001 | |
| Cubic curve | 0.991 | 189.186 | 3 | 5 | 0.000 | 12.180 | −0.298 | 0.004 | $-1.138 \times 10^{-5}$ |
| Compound curve | 0.775 | 24.146 | 1 | 7 | 0.002 | 3.428 | 1.006 | | |
| Power curve | 0.643 | 12.584 | 1 | 7 | 0.009 | 0.935 | 0.415 | | |
| S curve | 0.489 | 6.700 | 1 | 7 | 0.036 | 2.067 | −24.163 | | |
| Growth curve | 0.775 | 24.146 | 1 | 7 | 0.002 | 1.232 | 0.006 | | |
| Exponential curve | 0.775 | 24.146 | 1 | 7 | 0.002 | 3.428 | 0.006 | | |
| Logistic | 0.775 | 24.146 | 1 | 7 | 0.002 | 0.292 | 0.994 | | |

According to Figure 1 and Table 4, the best model of the fuel consumption curve for this model is a cubic curve model, which has the highest correlation, and its R-squared value is 0.991. Similarly, the curve fitting of the other four models of small vehicles by SPSS in the same way shows that the cubic curve model is still the most reliable. The R-squared values of model A, model C, model D and model E are 0.993, 0.978, 0.988 and 0.993, respectively, all of which have a very high correlation.

By observing the speed-fuel consumption curves of these five models, it can be seen that a vehicle speed of 60 km/h is an inflection point. When the vehicle speed is lower than 60 km/h, the fuel consumption decreases gradually with an increase in the vehicle speed. When the speed is higher than 60 km/h, the fuel consumption of a vehicle increases at a faster rate with an increase in speed. It can be seen that the most economic speed in this model is 60 km/h.

The parameters of the cubic curve model can be obtained from the model parameter table to establish the model equation of the model.

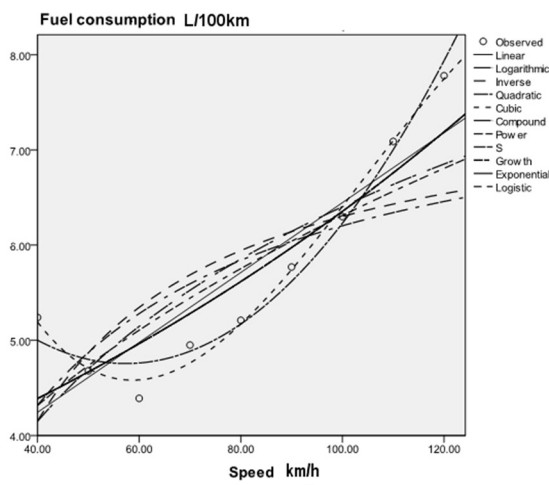

**Figure 1.** Fitting curves of model B.

Suppose the equation of a cubic curve model is:

$$F = b_3 v^3 + b_2 v^2 + b_1 v + c \tag{1}$$

where *F* is the fuel consumption value, with units of L/100 km; *V* is the travel speed, with units of km/h; and $b_3$, $b_2$, $b_1$, and c are values in the model parameter table. As shown in Table 4, the model equation of model B is:

$$F = 10^{-5} v^3 + 0.004 v^2 - 0.298 v + 12.180 \tag{2}$$

Similarly, according to the above method, the model equations of the other four models can be obtained as follows:

Model A:

$$F = 10^{-6} v^3 + 0.002 v^2 - 0.178 v + 9.257 \tag{3}$$

Model C:

$$F = 10^{-6} v^3 + 0.002 v^2 - 0.195 v + 10.614 \tag{4}$$

Model D:

$$F = 10^{-5} v^3 + 0.004 v^2 - 0.349 v + 12.558 \tag{5}$$

Model E:

$$F = 10^{-5} v^3 + 0.003 v^2 - 0.302 v + 13.111 \tag{6}$$

### 4.2. Fuel Consumption Model for Compact Vehicles

#### 4.2.1. Fuel Consumption Data Collection for Compact Vehicles

Compact vehicles, namely, class A cars, are the most common family cars, with a wheelbase generally between 2.5 m and 2.7 m and an engine displacement generally approximately 1.6 L to 2.0 L. Domestically, compact cars include own-brand models, joint venture models and pure imports. Data of the constant-speed fuel consumption of compact vehicles were collected from the Test of Easy Car platform, as shown in Table 5.

**Table 5.** Constant-speed fuel consumption of compact vehicles (L/100 km).

| Speed (km/h) | 40 | 50 | 60 | 70 | 80 | 90 | 100 | 110 | 120 |
|---|---|---|---|---|---|---|---|---|---|
| Z | 5.57 | 4.84 | 4.99 | 5.23 | 5.76 | 6.33 | 7.00 | 7.58 | 7.81 |
| G | 5.07 | 4.69 | 4.60 | 4.96 | 5.53 | 6.02 | 6.67 | 7.35 | 8.13 |
| H | 5.31 | 4.73 | 4.41 | 4.53 | 4.84 | 5.21 | 5.87 | 6.42 | 7.15 |
| I | 4.69 | 4.01 | 4.30 | 4.70 | 5.18 | 5.62 | 6.17 | 6.76 | 7.70 |
| J | 5.89 | 5.59 | 5.58 | 5.18 | 5.64 | 6.04 | 6.73 | 7.30 | 8.11 |

### 4.2.2. Fitting and Analysis of Fuel Consumption Data of Compact Vehicles

The fuel consumption data of compact vehicles were input into SPSS software for data analysis, and the fitting curves and model parameter summary table shown below were obtained.

Taking model J among the compact vehicles as an example, the model parameter summary table and fitting curves are shown in Table 6 and Figure 2.

**Table 6.** Model summary and parameter estimation of vehicle J.

| Model Equations | Model Summary | | | | | Parameter Estimation | | | |
|---|---|---|---|---|---|---|---|---|---|
| | R-Squared | F | $df_1$ | $df_2$ | Sig. | Constant | $b_1$ | $b_2$ | $b_3$ |
| Linear equation | 0.382 | 4.320 | 1 | 7 | 0.076 | 4.584 | 0.022 | | |
| Logarithmic curve | 0.242 | 2.230 | 1 | 7 | 0.179 | 0.765 | 1.289 | | |
| Reciprocal curve | 0.123 | 0.986 | 1 | 7 | 0.354 | 7.208 | −61.542 | | |
| Conic | 0.939 | 46.421 | 2 | 6 | 0.000 | 11.298 | −0.165 | 0.001 | |
| Cubic curve | 0.981 | 83.846 | 3 | 5 | 0.000 | 17.478 | −0.432 | 0.005 | $-1.480 \times 10^{-5}$ |
| Compound curve | 0.365 | 4.028 | 1 | 7 | 0.085 | 4.819 | 1.003 | | |
| Power curve | 0.229 | 2.073 | 1 | 7 | 0.193 | 2.727 | 0.193 | | |
| S curve | 0.113 | 0.895 | 1 | 7 | 0.376 | 1.965 | −9.070 | | |
| Growth curve | 0.365 | 4.028 | 1 | 7 | 0.085 | 1.573 | 0.003 | | |
| Exponential curve | 0.365 | 4.028 | 1 | 7 | 0.085 | 4.819 | 0.003 | | |
| Logistic | 0.365 | 4.028 | 1 | 7 | 0.085 | 0.208 | 0.997 | | |

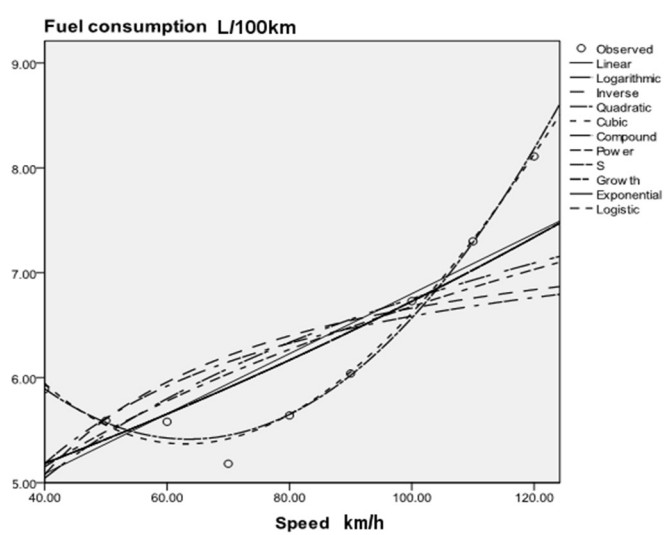

**Figure 2.** Fitting curves of model J.

According to the observed curve, a vehicle speed of 70 km/h is the turning point of the change curve of fuel consumption in the process of driving. When the vehicle speed is lower than 70 km/h, the fuel consumption decreases with an increase in vehicle speed. When the vehicle speed is higher than 70 km/h, the fuel consumption increases with an increase in the vehicle speed. This indicates that the most economic speed of the vehicle is 70 km/h. The model parameter table shows that the R-squared value of the cubic curve model is 0.981, which is far greater than those of the other models and slightly more than that of the quadratic curve model, so the cubic curve model is the most appropriate.

According to the above calculation formula, the model equation of model J can be obtained as follows:

$$F = 10^{-5}v^3 + 0.005v^2 - 0.432v + 17.478 \tag{7}$$

Similarly, through curve fitting in the same way, the R-squared values of the cubic curve models of the other 4 models can be obtained. The R-squared values of model Z, model G, model H and model I are 0.975, 0.997, 0.998 and 0.984, respectively. By the same method, the respective model equations can be obtained as follows:

Model Z:

$$F = 10^{-5}v^3 + 0.008v^2 - 0.589v + 18.487 \tag{8}$$

Model G:

$$F = 10^{-5}v^3 + 0.003v^2 - 0.258v + 10.925 \tag{9}$$

Model H:

$$F = 10^{-6}v^3 + 0.005v^2 - 0.285v + 12.282 \tag{10}$$

Model I:

$$F = 10^{-6}v^3 + 0.003v^2 - 0.209v + 9.234 \tag{11}$$

### 4.3. Fuel Consumption Model for Mid-Size Vehicles

#### 4.3.1. Fuel Consumption Data Collection for Mid-Size Vehicles

For a mid-size vehicle, the wheelbase is generally 2.7–2.9 m, the body length is generally 4.6–4.9 m, the engine displacement is generally approximately 2.0–3.0 L, and the price is approximately 200,000. Its body is relatively spacious, the appearance is atmospheric, and the price is relatively average. The constant-speed fuel consumption data of mid-size vehicles collected by the Test of Easy Car platform are shown in Table 7.

**Table 7.** Constant-speed fuel consumption of med-size vehicles (L/100 km).

| Speed (km/h) | 40 | 50 | 60 | 70 | 80 | 90 | 100 | 110 | 120 |
|:---:|:---:|:---:|:---:|:---:|:---:|:---:|:---:|:---:|:---:|
| K | 6.40 | 5.09 | 5.43 | 4.89 | 5.53 | 6.20 | 6.60 | 7.80 | 8.34 |
| L | 5.77 | 5.56 | 5.29 | 5.66 | 5.83 | 6.19 | 6.81 | 7.59 | 8.24 |
| M | 6.28 | 5.73 | 5.91 | 5.64 | 6.62 | 7.20 | 7.89 | 8.73 |
| N | 6.90 | 6.30 | 6.09 | 6.19 | 6.64 | 7.54 | 7.65 | 8.23 | 9.44 |
| O | 6.08 | 5.09 | 5.14 | 4.87 | 5.17 | 5.79 | 6.33 | 6.89 | 7.47 |

#### 4.3.2. Fitting and Analysis of Fuel Consumption Data of Mid-Size Vehicles

The fuel consumption data of 5 types of mid-size vehicles were also substituted into SPSS to obtain their respective model parameter tables and fitting curves. Model N is selected as an example, and the model parameter table and fitting curves are shown in Table 8 and Figure 3.

**Table 8.** Model summary and parameter estimation of vehicle N.

| Model Equations | Model Summary | | | | | Parameter Estimation | | | |
|:---:|:---:|:---:|:---:|:---:|:---:|:---:|:---:|:---:|:---:|
| | R-Squared | F | df$_1$ | df$_2$ | Sig. | Constant | b$_1$ | b$_2$ | b$_3$ |
| Linear equation | 0.705 | 16.722 | 1 | 7 | 0.005 | 4.497 | 0.034 | | |
| Logarithmic curve | 0.563 | 9.013 | 1 | 7 | 0.020 | −2.486 | 2.245 | | |
| Reciprocal curve | 0.412 | 4.901 | 1 | 7 | 0.062 | 9.029 | −128.236 | | |
| Conic | 0.958 | 67.758 | 2 | 6 | 0.000 | 9.654 | −0.110 | 0.001 | |
| Cubic curve | 0.969 | 51.482 | 3 | 5 | 0.000 | 13.302 | −0.267 | 0.003 | $-8.737 \times 10^{-6}$ |
| Compound curve | 0.704 | 16.664 | 1 | 7 | 0.005 | 4.982 | 1.005 | | |
| Power curve | 0.563 | 9.031 | 1 | 7 | 0.020 | 1.970 | 0.298 | | |
| S curve | 0.411 | 4.883 | 1 | 7 | 0.063 | 2.207 | −17.002 | | |
| Growth curve | 0.704 | 16.664 | 1 | 7 | 0.005 | 1.606 | 0.005 | | |
| Exponential curve | 0.704 | 16.664 | 1 | 7 | 0.005 | 4.982 | 0.005 | | |
| Logistic | 0.704 | 16.664 | 1 | 7 | 0.005 | 0.201 | 0.995 | | |

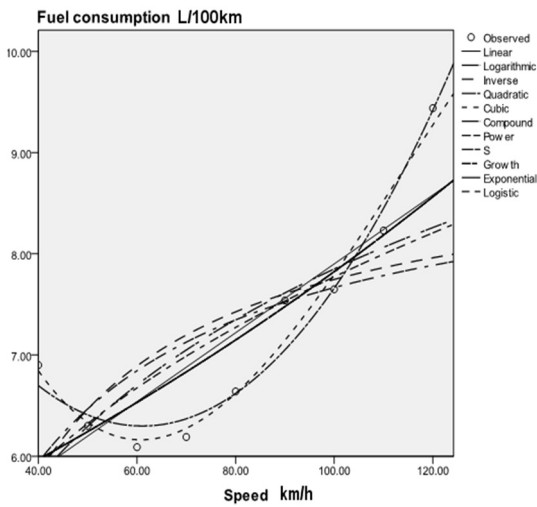

**Figure 3.** Fitting curves of model N.

The fitting curves show that the turning point is still located in the interval of 50–70 km/h. On the left side of the interval, the fuel consumption is inversely proportional to the speed, and on the right side of the interval, the fuel consumption is directly proportional to the speed.

The model parameter table shows that the R-squared value of the cubic curve model is 0.969, which is higher than those of the other models, indicating that its correlation coefficient is the highest. This model can be used as the best fuel consumption model.

The parameters of the cubic curve model can be obtained from the model parameter table to establish the model equation of a model N vehicle.

$$F = 10^{-6}v^3 + 0.003v^2 - 0.267v + 13.302 \tag{12}$$

Similarly, the cubic curve models of the other 4 models can be obtained. The R-squared values of model K, model L, model m and model O were 0.964, 0.994, 0.975 and 0.986, respectively. These results show that the correlation of the cubic curve model is the best. By the same method, the respective model equations can be obtained as follows:

Model K:
$$F = 10^{-5}v^3 + 0.005v^2 - 0.439v + 16.935 \tag{13}$$

Model L:
$$F = 10^{-6}v^3 + 0.002v^2 - 0.151v + 9.495 \tag{14}$$

Model M:
$$F = 10^{-6}v^3 + 0.002v^2 - 0.203v + 11.247 \tag{15}$$

Model O:
$$F = 10^{-5}v^3 + 0.004v^2 - 0.383v + 15.297 \tag{16}$$

*4.4. Fuel Consumption Model for SUV Vehicles*

4.4.1. Fuel Consumption Data Collection for SUV Vehicles

An SUV is a sport utility vehicle or suburban utility vehicle. Since they have strong power, have cross-country ability, are spacious and comfortable, and have the advantages of good cargo and passenger functions, SUVs, as urban new car preference models, have become the main automobile market growth in recent years, and now, SUVs generally refer to those vehicles based on a car platform to a certain extent, with the comfort of cars but also certain cross-country ability. The data of the

constant-speed fuel consumption of SUV models collected through the Test of Easy Car platform are shown in Table 9.

**Table 9.** Constant-speed fuel consumption of SUV vehicles (L/100 km).

| Speed (km/h) | 40 | 50 | 60 | 70 | 80 | 90 | 100 | 110 | 120 |
|---|---|---|---|---|---|---|---|---|---|
| P | 5.69 | 5.42 | 5.80 | 6.25 | 6.68 | 7.28 | 7.96 | 9.06 | 9.98 |
| Q | 5.68 | 5.54 | 5.73 | 6.24 | 6.80 | 7.13 | 7.92 | 9.07 | 9.72 |
| R | 8.03 | 7.65 | 6.98 | 7.65 | 7.85 | 8.75 | 9.77 | 10.67 | 11.55 |
| S | 7.08 | 6.63 | 6.19 | 6.88 | 7.65 | 8.54 | 9.50 | 10.70 | 11.98 |
| T | 7.33 | 7.26 | 6.80 | 7.15 | 8.00 | 8.90 | 10.08 | 10.80 | 11.82 |

### 4.4.2. Fitting and Analysis of Fuel Consumption Data of SUV Vehicles

The data in the above table were input into SPSS software for data analysis and curve fitting to obtain the optimal fuel consumption curve. Model T is selected as an example. The model parameter table and fitting curves are shown in Table 10 and Figure 4.

**Table 10.** Model summary and parameter estimation of vehicle T.

| Model Equations | Model Summary | | | | | Parameter Estimation | | | |
|---|---|---|---|---|---|---|---|---|---|
| | R-Squared | F | $df_1$ | $df_2$ | Sig. | Constant | $b_1$ | $b_2$ | $b_3$ |
| Linear equation | 0.855 | 41.156 | 1 | 7 | 0.000 | 3.764 | 0.061 | | |
| Logarithmic curve | 0.735 | 19.443 | 1 | 7 | 0.003 | −9.519 | 4.209 | | |
| Reciprocal curve | 0.591 | 10.123 | 1 | 7 | 0.015 | 12.239 | −252.091 | | |
| Conic | 0.977 | 125.319 | 2 | 6 | 0.000 | 9.641 | −0.103 | 0.001 | |
| Cubic curve | 0.992 | 215.368 | 3 | 5 | 0.000 | 16.781 | −0.411 | 0.005 | $-1.710 \times 10^{-5}$ |
| Compound curve | 0.860 | 42.844 | 1 | 7 | 0.000 | 4.927 | 1.007 | | |
| Power curve | 0.746 | 20.610 | 1 | 7 | 0.003 | 1.112 | 0.471 | | |
| S curve | 0.605 | 10.701 | 1 | 7 | 0.014 | 2.542 | −28.312 | | |
| Growth curve | 0.860 | 42.844 | 1 | 7 | 0.000 | 1.595 | 0.007 | | |
| Exponential curve | 0.860 | 42.844 | 1 | 7 | 0.000 | 4.927 | 0.007 | | |
| Logistic | 0.860 | 42.844 | 1 | 7 | 0.000 | 0.203 | 0.993 | | |

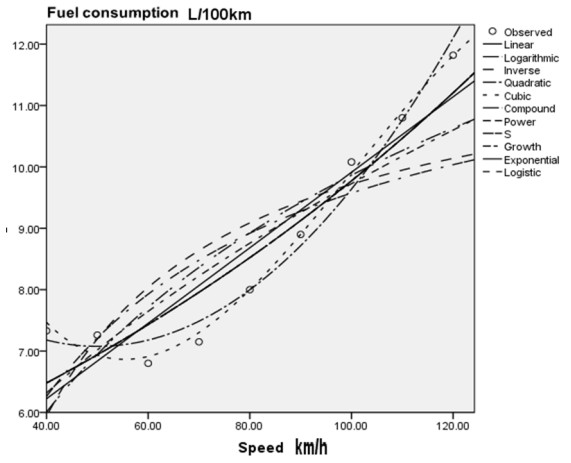

**Figure 4.** Fitting curves of model T.

According to the analysis of Table 10 and Figure 4, when the vehicle speed is 60 km/h, the fuel consumption curve shows an inflection point. When the speed is less than 60 km/h, the fuel consumption decreases with an increase in the speed; when the speed is more than 60 km/h, the fuel consumption increases with an increase in the speed.

The changes in the fuel consumption curve of model T are most consistent with the cubic curve model, and its R-squared value is 0.992. Similarly, the fuel consumption data of the other four models were imported into SPSS software for curve fitting, and the highest correlation coefficients of their respective fitting models were obtained for the cubic curve model.

The R-squared values of model P, model Q, model R and model S were 0.996, 0.993, 0.990 and 0.996, respectively. From the model summary and parameter estimation table, it can be seen that the cubic curve equations of the five models are as follows:

Model P:
$$F = 10^{-6}v^3 + 0.001v^2 - 0.101v + 7.763 \tag{17}$$

Model Q:
$$F = 10^{-6}v^3 + 0.002v^2 - 0.122v + 8.234 \tag{18}$$

Model R:
$$F = 10^{-5}v^3 + 0.005v^2 - 0.409v + 17.748 \tag{19}$$

Model S:
$$F = 10^{-5}v^3 + 0.005v^2 - 0.402v + 16.377 \tag{20}$$

Model T:
$$F = 10^{-5}v^3 + 0.005v^2 - 0.411v + 16.781 \tag{21}$$

## 5. Forecast and Analysis of Fuel Consumption for Different Models

### 5.1. Fuel Consumption Predictions of Different Models

According to the results of multi-model curve fitting for the fuel consumption data of different models by SPSS software mentioned above, the correlation coefficient of the cubic curve model is the largest, and the prediction model of fuel consumption about each vehicle model is established based on the fitting parameters. The predicted data of fuel consumption for small vehicles, compact vehicles, mid-size vehicles and SUVs are shown in Tables 11–14.

**Table 11.** Predicted fuel consumption of small vehicles (L/100 km).

| Speed (km/h) | 130 | 140 | 150 | 160 | 170 | 180 |
|---|---|---|---|---|---|---|
| A | 10.30 | 11.61 | 13.06 | 14.68 | 16.43 | 18.32 |
| B | 9.05 | 10.30 | 11.72 | 13.32 | 15.06 | 16.96 |
| C | 9.26 | 10.60 | 12.13 | 13.87 | 15.77 | 17.86 |
| D | 8.96 | 10.29 | 11.78 | 13.46 | 15.27 | 17.25 |
| E | 9.85 | 11.27 | 12.89 | 14.72 | 16.73 | 18.92 |

**Table 12.** Predicted fuel consumption of compact vehicles (L/100 km).

| Speed (km/h) | 130 | 140 | 150 | 160 | 170 | 180 |
|---|---|---|---|---|---|---|
| Z | 9.06 | 10.07 | 11.20 | 12.46 | 13.81 | 15.29 |
| G | 9.34 | 10.55 | 11.90 | 13.42 | 15.06 | 16.84 |
| H | 8.37 | 9.61 | 11.03 | 12.65 | 14.42 | 16.37 |
| I | 8.73 | 9.85 | 11.10 | 12.49 | 13.99 | 15.63 |
| J | 9.26 | 10.50 | 11.91 | 13.51 | 15.27 | 17.19 |

**Table 13.** Predicted fuel consumption of mid-size vehicles (L/100 km).

| Speed (km/h) | 130 | 140 | 150 | 160 | 170 | 180 |
|---|---|---|---|---|---|---|
| K | 10.03 | 11.63 | 13.46 | 15.56 | 17.85 | 20.38 |
| L | 9.35 | 10.49 | 11.79 | 13.26 | 14.86 | 16.60 |
| M | 9.92 | 11.19 | 12.63 | 14.26 | 16.03 | 17.98 |
| N | 10.58 | 11.91 | 13.41 | 15.11 | 16.98 | 19.02 |
| O | 8.80 | 10.06 | 11.52 | 13.18 | 15.01 | 17.02 |

**Table 14.** Predicted fuel consumption of SUV vehicles (L/100 km).

| Speed (km/h) | 130 | 140 | 150 | 160 | 170 | 180 |
|---|---|---|---|---|---|---|
| P | 11.24 | 12.58 | 14.07 | 15.72 | 17.49 | 19.41 |
| Q | 10.98 | 12.24 | 13.62 | 15.16 | 16.81 | 18.59 |
| R | 13.39 | 15.15 | 17.14 | 19.39 | 21.83 | 24.51 |
| S | 13.98 | 16.00 | 18.27 | 20.82 | 23.58 | 26.58 |
| T | 13.65 | 15.39 | 17.33 | 19.50 | 21.85 | 24.40 |

Curves of the fuel consumption changes for small vehicles, compact vehicles, mid-size vehicles and SUV models are shown in Figures 5–8.

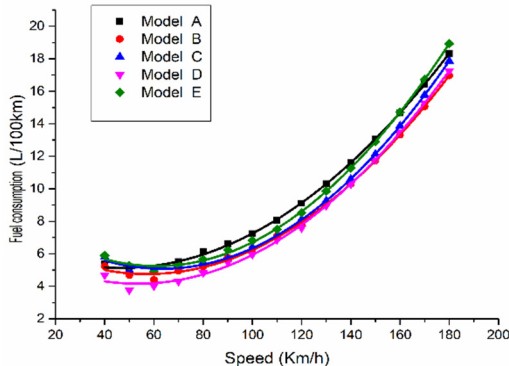

**Figure 5.** Fuel consumption of small vehicles.

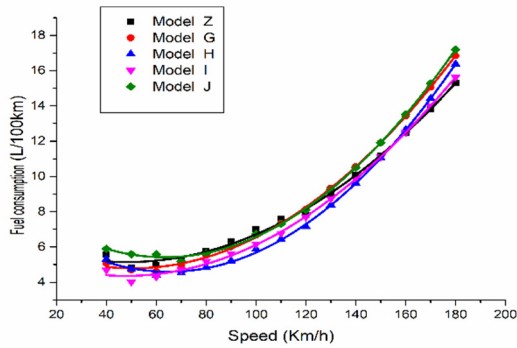

**Figure 6.** Fuel consumption of compact vehicles.

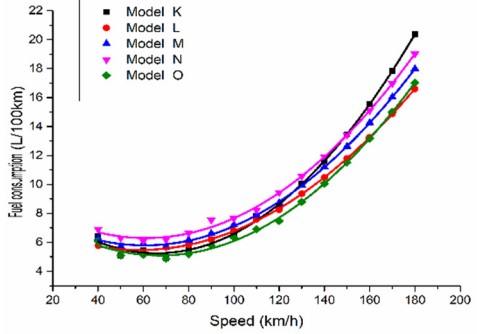

**Figure 7.** Fuel consumption of mid-size vehicles.

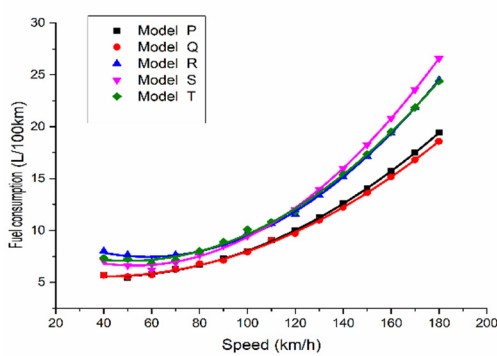

**Figure 8.** Fuel consumption of SUV vehicles.

It can be seen from the curves that the model with the highest fitting degree for the fuel consumption models is a cubic curve for small vehicles, compact vehicles, mid-size vehicles and SUVs. By comparing the R-squared values in the model summaries and parameter estimation tables obtained by SPSS software, the R-squared value of the cubic curve model is the largest, which also indicates that the correlation of the cubic curve is the strongest. This conclusion can be extended to the relationship between fuel consumption and speed for all models in line with the change in a cubic curve; similarly, the consumption of fuel of vehicles on a superhighway can also be studied by using a cubic curve model. In addition, by observing the slight changes in the curves, the economic speed ranges of various models are similar, most of which are located in the range of 50–70 km/h, and the fuel consumption also shows an increasing trend with an increase in speed. This shows the similarity of the prediction models of different vehicle models.

The performance of different automobile engines, vehicle loads, fuel types and tires are different, resulting in variation in different types of automobile fuel consumption levels. For the fuel consumption curves, the change rates of the curves, namely, the curvature, and the fuel consumption corresponding to the same speed are different, which also shows the differences in the fuel consumption models of different vehicle types.

In addition, the mean square error analysis is carried out for the above 20 models of fuel consumption prediction. It is found that the mean square error of the prediction models established for each vehicle type is less than 0.1, indicating a small deviation between the predicted fuel consumption value and the actual value. Therefore, the established fuel consumption prediction model has a high accuracy and can be used to predict vehicle fuel consumption with a speed over 120 km/h. The mean square error of fuel consumption prediction models of different vehicle types is shown in Table 15.

**Table 15.** Mean square error of fuel consumption prediction models of different vehicle types.

| Model | Mean Square Error | Model | Mean Square Error | Model | Mean Square Error | Model | Mean Square Error |
|-------|------------------|-------|------------------|-------|------------------|-------|------------------|
| A | 0.027 | Z | 0.085 | K | 0.071 | P | 0.014 |
| B | 0.046 | G | 0.030 | L | 0.011 | Q | 0.024 |
| C | 0.037 | H | 0.023 | m | 0.022 | R | 0.087 |
| D | 0.077 | I | 0.047 | N | 0.070 | S | 0.074 |
| E | 0.038 | J | 0.022 | O | 0.064 | T | 0.088 |

*5.2. Comparative Analysis of Fuel Consumption of Different Models*

The average forecast fuel consumption of different models is shown in Table 16. The fuel consumption of the simulation equations of the four models is shown in Figure 9.

As shown in Figure 9, when the speed is less than 120 km/h, the fuel consumption curves of small vehicles and compact vehicles are very close, indicating that the fuel consumption characteristics of small vehicles and compact vehicles are similar and it is basically the same. The consumption of fuel of mid-size vehicles and SUV models is the highest. When the speed exceeds 120 km/h, with increasing speed, the fuel consumption of small vehicles exceeds that of compact vehicles and is close to that

of mid-sized vehicles. At this point, the amount of fuel compact vehicles consume is the smallest, the fuel consumption of small vehicles and mid-sized vehicles is basically the same, and SUV models still consume the most fuel. These results show that the fuel consumption characteristics of a vehicle have a greater relationship with the model.

**Table 16.** Predicted average fuel consumption by vehicle type.

| Speed (km/h) | 130 | 140 | 150 | 160 | 170 | 180 |
|---|---|---|---|---|---|---|
| Small vehicles | 9.48 | 10.81 | 12.32 | 14.01 | 15.85 | 17.86 |
| Compact vehicles | 8.95 | 10.12 | 11.43 | 12.91 | 14.51 | 16.26 |
| Mid-size vehicles | 9.74 | 11.06 | 12.56 | 14.27 | 16.15 | 18.20 |
| SUV vehicles | 12.65 | 14.27 | 16.09 | 18.12 | 20.31 | 22.70 |

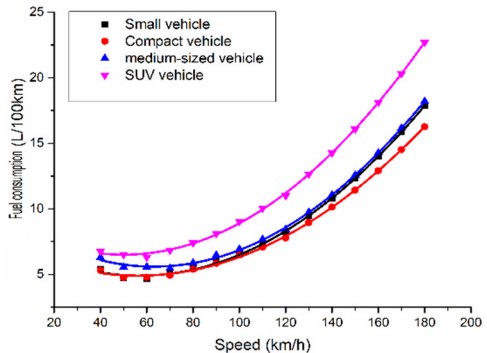

**Figure 9.** Fuel consumption of different models.

In addition, Origin software is used to carry out average fitting of the five sets of curves of the four models to obtain their respective fitting equations as follows:

Small vehicles:
$$F = 10^{-6}v^3 + 0.003v^2 - 0.245v + 11.008 \tag{22}$$

Compact vehicles:
$$F = 10^{-5}v^3 + 0.004v^2 - 0.305v + 12.560 \tag{23}$$

Mid-size vehicles:
$$F = 10^{-6}v^3 + 0.003v^2 - 0.276v + 12.947 \tag{24}$$

SUV vehicles:
$$F = 10^{-5}v^3 + 0.004v^2 - 0.293v + 13.524 \tag{25}$$

## 6. Conclusions

By fitting and analyzing the fuel consumption data of small vehicles, compact vehicles, mid-size vehicles and SUV models provided by the Test of Easy Car platform, the following conclusions are drawn:

(1) By fitting the fuel consumption data of different models provided by the Test of Easy Car platform, and the results showed that the R-squared values of the cubic curves were the largest; all of these values exceeded 0.9, and some even approached 1. These results show that a cubic curve has the best correlation with the trend of the variation in the fuel consumption curves, so cubic curve models of small vehicles, compact vehicles, mid-size vehicles and SUVs are established.

(2) Through fitting and analyzing the fuel consumption data of a total of 20 models of the 4 categories. The results showed that regardless of the model, the rule of the variation in the fuel consumption curves was similar; that is, all models had an economic speed range, and the economic speed range was between 50–70 km/h.

(3) The estimated values of fitting parameters were extracted by the software, and fuel consumption models of different types of vehicles were established. The models were used to predict fuel consumption at different speeds on a superhighway, and fuel consumption curves of different models were drawn. According to the change curves of fuel consumption, when the vehicle speed is higher than the economic speed, the fuel consumption starts to increase, and the rate of increasing it increases with an increase in the speed.

(4) Through the established fuel consumption prediction model, it can be found that the models with the least and the most fuel consumption are compact vehicles and SUV vehicles respectively. This can provide a reference for road users to choose the most economical and fuel-efficient vehicle when buying a car. When they drive vehicles with the lowest fuel consumption on the roads, it will not only save money on fuel consumption, but also reduce vehicle exhaust emissions due to fuel consumption. In addition, the prediction model is used to predict vehicle fuel consumption when the speed exceeds 120 km/h, and it can also be used to estimate the cost of vehicles running on the superhighway with the design speed exceeding 120 km/h, so as to provide a basis for demonstrating the feasibility of the superhighway.

Some European and American scholars have also studied automobile energy consumption in many aspects, such as the relationship between engine performance, driving strategy, engine control and vehicle fuel consumption. The results show that the nonlinear model is the most suitable for the variation law of vehicle fuel consumption. In addition, when the speed exceeds 120 km/h, the fuel consumption of the vehicle increases with the increase of the speed. The above research results are basically consistent with our team's previous predictions, but due to the single source of data, the research results need to be further tested. Our research team will optimize the fuel consumption model for superhighways by means of an engine bench test and real vehicle road test. In addition, the controlled variable in this study is speed, the correlation between speed and fuel consumption is considered, and the prediction model of fuel consumption and speed is established. In the next step, factors such as different road conditions, different vehicle characteristics and different driving styles will be considered to influence the fuel consumption of vehicles on expressway, so as to optimize the established fuel consumption prediction model.

**Author Contributions:** Conceptualization, Y.-M.H., Y.-L.P. and J.K.; data curation, J.K. and Y.-T.S.; Formal analysis, Y.-M.H., J.K. and Y.-T.S.; Funding acquisition, Y.-M.H. and Y.-L.P.; Investigation, Y.-M.H., J.K. and Y.-T.S.; Methodology, B.R. and Y.-L.P.; Project administration, Y.-M.H. and J.K.; Supervision, B.R. and Y.-L.P.; Validation, Y.-M.H., B.R. and Y.-L.P.; Writing—Original draft, Y.-M.H. and J.K.; Writing—Review & Editing, B.R. and Y.-L.P. All authors have read and agreed to the published version of the manuscript.

**Funding:** We are thankful to the Committee of Natural Foundation of Heilongjiang Province, China, for providing financial support with Project of Natural (LH2019E004); thankful to Education Department of Heilongjiang Province, China, for providing financial support with Key Project of Education Science Planning of Heilongjiang Province (No HLJ20191010). Furthermore, this research was also partially supported by Northeast Forestry University, with College Reform Project (DL 20190005). The study reflects only the authors' views and the Committee of Natural Foundation of Heilongjiang Province, the Education Department of Heilongjiang Province and Northeast Forestry University are not liable for any use that may be made of the information contained herein.

**Acknowledgments:** Thanks to Shao-Yan Li, librarian of Northeast Forestry University, for providing helps in consulting electronic periodicals and books. Thanks to Yang Ronghai, Office Director of Northeast Forestry University. Thanks to Yang Cheng, professor of the College of Engineering, University of Wisconsin-Madison, for his suggestions on the structure of the paper.

**Conflicts of Interest:** The authors declare no conflict of interest.

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
