# Peer review of "Study on a Prediction Model of Superhighway Fuel Consumption Based on the Test of Easy Car Platform"

_sustainability, doi:10.3390/su12156260_

Round 1

Reviewer 1 Report

The reviewed article is another article on the functioning of the planned super highway in China (with a speed limit of up to 180 km / h). This time the authors present the problem of fuel consumption of passenger cars when driving at high speeds.
After reading and analyzing the article, the Reviewer presents the following Notes to individual parts of the article:
0. "Abstract" should contain: a brief description of the research problem, general purpose of the research, characteristics of the training ground and research method as well as general results. However, after outlining the problem, the authors unnecessarily presented detailed research results, which makes reading the work difficult.
1. "Introduction" should contain a detailed description of the problem, genesis and specific objectives of the studies and analyzes carried out. The authors practically repeated the abstract of the thesis, but did not pose research questions, which specify the purpose of the work and the basis for subsequent discussion of the research results.
2. "Overview of Fuel Consumption Research on Superhighways" - contains mainly a description of the authors' research on the design of a super highway designed in China, and they have very little research into the fuel consumption of cars and their modeling. This chapter should refer to literature studies at the review of national and world studies (recommended proportion of the cited literature: 10% authors, 40% country of origin of authors, 50% rest of the world). There is a lack of literature in this review:
a. description and critical analysis of research carried out by global research teams, among others from Japan, the USA, Germany and Sweden;
b. description of the research methods used,
c. description of the set of the most important factors (vehicle design, road geometry, pavement condition, speed limit, driving style, etc.) affecting fuel consumption
d. description of the results and generalizations in the form of selected models of fuel consumption (FC) from the set of the most important factors and models of passenger car (PE) efficiency, in particular models for vehicles traveling at high speed (close to 200 km / h).
e. The methods used to forecast the fuel consumption of passenger cars.
3. "Collection of Fuel Consumption Data" contains a general description of the acquisition of fuel consumption data for selected passenger cars in the speed range 40 - 120 km / h. Missing in this chapter:
a. Specifying the testing ground (road and its geometry),
b. Driving style and aerodynamic characteristics, weight and other vehicle characteristics),
c. Methods of obtaining data on the fuel consumption of individual vehicles and analysis of the accuracy of measuring fuel consumption,
d. Distributions of data sets in individual speed classes together with their parameters.
4. "Data Analysis and Curve Fitting by SPSS Software" - reservations are raised by:
a. Approving the models of the dependence of the amount of fuel consumption (FC) on the speed based on a limited number of measurements (9 results for each function),
b. The use of the R regression coefficient to assess the quality of matching the adopted function to the data (very high R coefficients were obtained due to the small amount of data),
c. For curvilinear functions, additional measures of function matching should be used (e.g. mean square error),
d. There are no variables other than speed that affect the amount of fuel consumed by vehicles.

e. Too much detailed data, results should be presented in a more general form.

5. "Forecast and Analysis of Fuel Consumption for Different Models" - reservations arise:
a. Adoption of the fuel consumption model developed on the basis of a limited number of data (9 data) and a limited speed range (40 - 120 km) for extrapolation of data to fuel consumption models for vehicles developing speeds up to 180 km / h,
b. Lack of analysis of fuel consumption prediction errors when using of the cubic curves models, these models can cause quite large errors at high speeds.
c. Lack of verification and validation of proposed predictive models. Proposed predictive models can be treated as preliminary to proper research.
6. "Conclusion" - the conclusions are a summary of the research, but the following are missing:
a. Discussion of the results obtained and models of fuel consumption by motor vehicles, in particular against the background of researchers from other countries.
b. A program of further research allowing for credibility of the proposed models and taking into account the determination of the impact of other factors on the fuel consumption of vehicles on super highways in China.

Author Response

Response to Reviewer 1 Comments

The reviewed article is another article on the functioning of the planned super highway in China (with a speed limit of up to 180 km / h). This time the authors present the problem of fuel consumption of passenger cars when driving at high speeds.
After reading and analyzing the article, the Reviewer presents the following Notes to individual parts of the article:

Point 0: "Abstract" should contain: a brief description of the research problem, general purpose of the research, characteristics of the training ground and research method as well as general results. However, after outlining the problem, the authors unnecessarily presented detailed research results, which makes reading the work difficult.

Response 0: Thank you very much for your suggestions. I have revised the detailed research results in the manuscript to discuss the scope instead. The fuel consumption prediction performance difference of different vehicle models with a speed of 130km/h~180km/h is discussed.

Point 1: "Introduction" should contain a detailed description of the problem, genesis and specific objectives of the studies and analyzes carried out. The authors practically repeated the abstract of the thesis, but did not pose research questions, which specify the purpose of the work and the basis for subsequent discussion of the research results.

Response 1: I am sorry that I did not elaborate on the research questions and other contents in the introduction. On the one hand, through establishing fuel consumption prediction models of different vehicle types and comparing the differences in fuel consumption performance of different vehicle types, this study can provide economic reference for road users when they choose different vehicle types. In addition, the established fuel consumption prediction model can also provide reference for the fuel consumption prediction of vehicles on the superhighway with a design speed over 120km/h, and estimate the operating cost of vehicles on the superhighway. Finally, the difference in fuel consumption performance is also related to vehicle exhaust emissions, which can be used as a reference for studying exhaust emissions of different vehicle models. I have added this to the introduction in the manuscript.

Point 2: "Overview of Fuel Consumption Research on Superhighways" - contains mainly a description of the authors' research on the design of a super highway designed in China, and they have very little research into the fuel consumption of cars and their modeling. This chapter should refer to literature studies at the review of national and world studies (recommended proportion of the cited literature: 10% authors, 40% country of origin of authors, 50% rest of the world). There is a lack of literature in this review:
a. description and critical analysis of research carried out by global research teams, among others from Japan, the USA, Germany and Sweden;
b. description of the research methods used,
c. description of the set of the most important factors (vehicle design, road geometry, pavement condition, speed limit, driving style, etc.) affecting fuel consumption
d. description of the results and generalizations in the form of selected models of fuel consumption (FC) from the set of the most important factors and models of passenger car (PE) efficiency, in particular models for vehicles traveling at high speed (close to 200 km / h).
e. The methods used to forecast the fuel consumption of passenger cars.

Response 2: Thank you very much for your advice. In section 2.2, I have explained the literature review on vehicle fuel consumption at home and abroad. However, there are some shortcomings, so I have added the corresponding literature and its content description in section 2.2 of the manuscript according to your suggestion.

  1. I have described the relevant studies on foreign fuel consumption in Section 2.2.
  2. In this study, the fuel consumption data of different vehicle types at a speed of 40~120km/h were analyzed, and the curve fitting was carried out by SPSS, the data analysis software, to obtain the fitting model with the greatest correlation. Then the difference of fuel consumption performance of different models is analyzed and predicted respectively. The variation law of fuel consumption of different vehicle models with a speed of more than 120km/h is discussed.

c.Vehicle design, road geometry, road conditions, speed limit, driving mode and other important factors have a great impact on vehicle fuel consumption. For example, the design of vehicle shape and tires will make the vehicle subject to different degrees of air resistance and rolling resistance during driving, which will consume the energy generated by vehicle fuel. When the road conditions are good, the work done by the car to overcome the road resistance will be reduced, which in turn will reduce the fuel consumption. Under the influence of engine performance, road conditions, load, wind direction, climate and usage, the vehicle will consume the least amount of fuel when driving within a certain speed range, which is the economic speed range. Generally speaking, the geometry of different roads will also have an impact on the fuel consumption of vehicle. When there are more uphill sections and turning sections, the vehicle must overcome the uphill resistance or brake to decelerate the turning to achieve the driving purpose, which will increase fuel consumption to a certain extent.

  1. The driving speed of vehicles has an important impact on vehicle fuel consumption. Different models will have a certain impact on vehicle fuel consumption due to engine performance, vehicle design and other factors. The study of the fuel consumption law of different passenger car models and the establishment of relevant models can provide a reference for road users to choose the most economical and fuel-efficient car models. In addition, vehicle exhaust emissions are closely related to fuel consumption, and the results of this study can also provide reference for the study of exhaust emissions of different vehicle models.
  2. In addition, there are many methods to predict the fuel consumption of passenger vehicles, such as artificial neural network method, multiple linear regression, least square method and simulation modeling method, etc. However, different methods have different advantages and disadvantages, which must be selected according to different vehicle types, different road conditions, and different driving modes, etc. Most importantly, the prediction results should be verified and error analyzed, and necessary corrections should be made to ensure the accuracy of the prediction results.

Point 3: "Collection of Fuel Consumption Data" contains a general description of the acquisition of fuel consumption data for selected passenger cars in the speed range 40 - 120 km / h. Missing in this chapter:
a. Specifying the testing ground (road and its geometry),
b. Driving style and aerodynamic characteristics, weight and other vehicle characteristics),
c. Methods of obtaining data on the fuel consumption of individual vehicles and analysis of the accuracy of measuring fuel consumption,
d. Distributions of data sets in individual speed classes together with their parameters.

Response 3:

  1. The test site is a flat road, mentioned in line 203.The fuel consumption data selected in this paper are from the Test of Easy Car platform Website, a Chinese automobile testing website. It conducts professional tests on the power, economy and safety of vehicles sold on the market, and publishes the results on its official website for consumers' reference. This is mentioned in line 54 of the introduction.
  2. The influence of driving style and aerodynamic characteristics on vehicle fuel consumption is uncontrollable, and the final fuel consumption test results have little reference value for consumers. On the other hand, constant speed fuel consumption is different, because the test method for each vehicle under test is fixed and highly repeatable. Consumers can compare the fuel consumption results at each speed level through constant speed fuel consumption test to find out the difference in fuel consumption of vehicles of the same grade and displacement at the same speed.

The different models selected are mainly small cars, compact cars, medium cars and SUV cars. The focus of this study is to discuss the law and performance of fuel consumption of different models and establish fuel consumption prediction models of different models, so there is not much introduction to vehicle characteristics.

  1. first we used is one percent ml measuring precision of fuel consumption instrument to fuel consumption test data collection, in the test, we will fuel consumption instrument meter series connection on the way vehicle engine to be tested for tubing, supply the combustion of gasoline engine all through our fuel consumption instrument for measurement, then the fuel flow rate of fuel consumption instrument measurement is the actual fuel consumption of engine."Easy car test" adopted by the connection method to test the fuel consumption meter, can fundamentally solve the problems of fuel consumption measurement is allowed to jump shot, and considering the physical properties of objects "heat bilges cold shrink", we adopt the national standard correction formula is given for the final data for correction of fuel consumption test, objective and accurate fuel consumption test results are obtained. By these means, we will minimize the error of fuel consumption test results. This part has been mentioned in section 3.1 of the manuscript.
  2. Detailed data of fuel consumption are shown in Table 3, Table 5, Table 7 and Table 9.

Point 4: "Data Analysis and Curve Fitting by SPSS Software" - reservations are raised by:
a. Approving the models of the dependence of the amount of fuel consumption (FC) on the speed based on a limited number of measurements (9 results for each function),
b. The use of the R regression coefficient to assess the quality of matching the adopted function to the data (very high R coefficients were obtained due to the small amount of data),
c. For curvilinear functions, additional measures of function matching should be used (e.g. mean square error),
d. There are no variables other than speed that affect the amount of fuel consumed by vehicles.

  1. Too much detailed data, results should be presented in a more general form.

Response 4:

  1. There is a close relationship between fuel consumption and speed. Based on the data fitting and modeling of 9 fuel consumption data, the great correlation between fuel consumption and speed is shown to a certain extent.
  2. You are right. Therefore, I selected a variety of vehicle models for research and found that the law of fuel consumption change with speed is similar.
  3. When SPSS was used for curve fitting, the sig values in the output model summary and parameter estimation table (Table 4, Table 6, Table 8 and Table 10) were all less than 0.05, indicating a high level of significance and statistically significant.
  4. It is true that other variables will affect the fuel consumption of vehicles, but the fuel consumption data tested in this study were tested in the same method and under the same conditions, and only one variable was controlled, namely speed. The purpose is to explore the relationship between speed and fuel consumption and establish a vehicle fuel consumption prediction model with a speed over 120km/h.
  5. The detailed data allows the entire modeling process to be reproduced for the reader's understanding. The general results can be referred to Figure 5~ Figure 8 in Chapter 5.

Point 5: "Forecast and Analysis of Fuel Consumption for Different Models" - reservations arise:
a. Adoption of the fuel consumption model developed on the basis of a limited number of data (9 data) and a limited speed range (40 - 120 km) for extrapolation of data to fuel consumption models for vehicles developing speeds up to 180 km / h,
b. Lack of analysis of fuel consumption prediction errors when using of the cubic curves models, these models can cause quite large errors at high speeds.
c. Lack of verification and validation of proposed predictive models. Proposed predictive models can be treated as preliminary to proper research.

Response 5:

  1. According to Chinese law, expressway speed limit is 120km/h, and speeding is severely punished, so the speed range cannot exceed 120km/h. Through curve fitting, it is found that the limited data are closely related to each other, which all conform to the law of cubic curve change. The cubic curve model established can be used as the basis for appropriate research.
  2. and c. Due to the limitation of conditions and Chinese law, vehicles are not allowed to travel on highways at speeds exceeding 120km/h. Therefore, it is impossible to carry out the actual vehicle experiment and obtain the vehicle fuel consumption value with a speed of 130~180km/h, so as to verify with the prediction model. However, in 2018, construction began on China's first Hangzhou-Shaoyung expressway with a speed limit of 150km/h, which will be open to traffic by 2020.When the highway is built, if conditions permit, it can be carried out on the actual vehicle road experiment and verified with the established fuel consumption prediction model.

Point 6: "Conclusion" - the conclusions are a summary of the research, but the following are missing:
a. Discussion of the results obtained and models of fuel consumption by motor vehicles, in particular against the background of researchers from other countries.
b. A program of further research allowing for credibility of the proposed models and taking into account the determination of the impact of other factors on the fuel consumption of vehicles on super highways in China.

Response 6:

  1. The test method of fuel consumption on the Test of Easy Car platform is same as the real car experiment, the test vehicle is also the vehicle sold on the market, combined with the actual. Some European and American scholars have also studied automobile energy consumption in many aspects, such as the relationship between engine performance, driving strategy, engine control and vehicle fuel consumption. The results show that the nonlinear model is the most suitable for the variation law of vehicle fuel consumption. In addition, when the speed exceeds 120km/h, the fuel consumption of the vehicle increases with the increase of the speed.
  2. The next section will consider the influences of different road conditions, different vehicle characteristics and different driving styles on the fuel consumption of expressway vehicles. I have added this to the conclusion.

Reviewer 2 Report

Reviewed paper has good structure. Authors prepared enough literature review. Topic of the paper is interesting and important. Results may be useful for some readers. What more, in my opinion, description of whole modelling process from the paper may be also useful for younger researchers. They will find here some kind of giude - how to prepare assumptions, collect data and realize model.

Figures have good quality.

Authors also described further plans related to this research area.

Based on above information, in my opinion the paper is ready to publish.

Author Response

Response to Reviewer 2 Comments

Comments:

Reviewed paper has good structure. Authors prepared enough literature review. Topic of the paper is interesting and important. Results may be useful for some readers. What more, in my opinion, description of whole modelling process from the paper may be also useful for younger researchers. They will find here some kind of giude - how to prepare assumptions, collect data and realize model.

Figures have good quality.

Authors also described further plans related to this research area.

Based on above information, in my opinion the paper is ready to publish.

Response:

First of all, thank you for taking the time to read and comment on my article. In addition, I would like to thank you for your recognition of my research results.

Reviewer 3 Report

Before making any comments on the contents of this manuscript, I would like to share a few opinions on the style and writing of the text. I have not encountered huge mistakes, but truth to be told I think the text is slightly “heavy” in some sections. Many words and expressions are consistently repeated within short sections, thus giving the manuscript a very slow pace. For instance, see this section from part 2.: “In 2016, our research team studied the different stages […]., and conducted an in-depth study […]. In 2017, our research team analyzed the driving characteristics […], conducted an in-depth study […]. In 2018, our research team conducted a comparative study on […]. Etc” I am aware that what matters the most are the contents of the article, but I believe that checking the style in which these are conveyed can really make a difference in how readers receive them. Using some synonyms and trying not to stick to a very repetitive way of writing could improve greatly the style of your work, so I strongly suggest checking it again and fixing any possible repetition.

Also, be careful with typos and editing mistakes. For instance, in line 131: “Graf von Westarp, A found that the parameters of many existing fuel consumption […]”. What is this “A” suppose to mean? Do you mean “has found”, or is this an editing typo?

Now, moving on with the contents:

Abstract: I do not consider it very appropriate to include all values of kilometers, and neither of liters per kilometers, I think that it would be more adequate to discuss ranges in this section.

Please state what is “SUVs”, which I imagine refers to sport utility vehicles; you are using it without any previous mention, both in the abstract and at the introduction. However, it is not only until section 4.4.1 that you refer to it as “is a sport utility vehicle or suburban utility vehicle”, with no reference.

2.1 Research on Superhighways: I think the case of France should be included, where roads with 130km/h speed have existed for at least 15 years, and, taking into account elements related to climate change as well, they are considering reducing these speed limits. Additionally, you should talk about accident rates and environmental effects, such as rain, fog etc.

4.1.1 Fuel consumption data collection for small vehicles: please include a definition of the Test of Easy Car platform, how it was implemented, who developed it, etc., either in line 54 of the introduction, or in this section. What must be in this section is the explanation of how the data collected through this platform were accessed (and the ethical requirements for the data protection, if any).

Discussion: this work has no discussion, and it must be included. Answer questions such as: how could these findings generate a contribution to road designs? Which are the implications of these results for road safety, for the economy, for global climate change, for a road user to choose one car instead of another?

Author Response

Response to Reviewer 3 Comments

Point 1: Before making any comments on the contents of this manuscript, I would like to share a few opinions on the style and writing of the text. I have not encountered huge mistakes, but truth to be told I think the text is slightly “heavy” in some sections. Many words and expressions are consistently repeated within short sections, thus giving the manuscript a very slow pace. For instance, see this section from part 2.: “In 2016, our research team studied the different stages […]., and conducted an in-depth study […]. In 2017, our research team analyzed the driving characteristics […], conducted an in-depth study […]. In 2018, our research team conducted a comparative study on […]. Etc” I am aware that what matters the most are the contents of the article, but I believe that checking the style in which these are conveyed can really make a difference in how readers receive them. Using some synonyms and trying not to stick to a very repetitive way of writing could improve greatly the style of your work, so I strongly suggest checking it again and fixing any possible repetition.

Also, be careful with typos and editing mistakes. For instance, in line 131: “Graf von Westarp, A found that the parameters of many existing fuel consumption […]”. What is this “A” suppose to mean? Do you mean “has found”, or is this an editing typo?

Response 1: Thank you very much for your good advice. I am sorry that I did not notice in the writing that many words and expressions are repeated in the short paragraphs. I have carefully checked and revised the corresponding parts in the manuscript. Also, in line 131, "Graf von Westarp, A" is my edit error, which is the name of an author. I have changed it to "Graf von Westarp A" in the manuscript.

Point 2: Abstract: I do not consider it very appropriate to include all values of kilometers, and neither of liters per kilometers, I think that it would be more adequate to discuss ranges in this section.

Response 2: Thank you very much for your advice. I have listed all the data just to show the accuracy of fuel consumption forecast data, so as to show the trend of fuel consumption forecast more. However, your suggestion is good and the discussion scope is sufficient to indicate the trend of fuel consumption prediction, and the accurate value of fuel consumption prediction is clearly given in Table 11-15 in the body. So I have corrected it in the abstract of the manuscript.

Point 3: Please state what is “SUVs”, which I imagine refers to sport utility vehicles; you are using it without any previous mention, both in the abstract and at the introduction. However, it is not only until section 4.4.1 that you refer to it as “is a sport utility vehicle or suburban utility vehicle”, with no reference.

Response 3: I am very sorry that this is my negligence , the abbreviation of sport utility vehicle is "SUV" in China, which caused me to forget the explanation when I used it for the first time. I have explained this in the abstract of the manuscript.

Point 4: 2.1 Research on Superhighways: I think the case of France should be included, where roads with 130km/h speed have existed for at least 15 years, and, taking into account elements related to climate change as well, they are considering reducing these speed limits. Additionally, you should talk about accident rates and environmental effects, such as rain, fog etc.

Response 4: Thank you very much for your advice. In an effort to improve the climate, the French Citizens' Climate Conference has proposed lowering the speed limit on motorways to 110km/h. However, doing so is only a drop in the ocean and cannot fundamentally solve the climate problem. The speed limit of 130km/h on French motorways has existed for 15 years, and most drivers have been accustomed to such speed. If the speed is reduced prematurely, it will have a great impact on citizens' travel and other activities. In addition, with the rapid development of highway construction technology and automobile technology, autonomous driving technology has become increasingly mature and gradually industrialized. From Level 0 to Level 5, human intervention is less and less, the degree of intelligence is higher and higher, and the safety is more and more guaranteed. With the improvement of the level of automatic driving, safety hazards such as delayed reaction and wrong judgment caused by human physiological factors are less and less, or even eliminated, which greatly reduces the threat of adverse environment (such as rain and snow) to driving safety. Besides, Level 4 self-driving cars have entered the market, which can effectively guarantee the operation safety of superhighways. Therefore, from the perspective of highway construction technology, vehicle performance and foreign experience, conditions are ripe for the construction of superhighways with design speed exceeding 120km/h in China.

Point 5: 4.1.1 Fuel consumption data collection for small vehicles: please include a definition of the Test of Easy Car platform, how it was implemented, who developed it, etc., either in line 54 of the introduction, or in this section. What must be in this section is the explanation of how the data collected through this platform were accessed (and the ethical requirements for the data protection, if any).

Response 5: The Test of Easy Car Platform is an automobile testing website in China. It specializes in performing professional tests on the power, economy and safety of vehicles sold on the market, and publishes the results on its official website for consumers' reference. Its test data is public and accessible via the Internet, which may be inaccessible abroad because of Internet blockades. In addition, led by Guo Konghui, academician of the Chinese Academy of Engineering, 13 senior experts in the automobile industry have formed the expert Committee of "Easy Vehicle Test". The committee will greatly enhance the professional level and credibility of the "easy vehicle test" project, and provide the most reliable and effective guarantee for the professionalism and accuracy of vehicle testing. The details have been added in line 54 of the introduction and in Chapter 3.

Point 6: Discussion: this work has no discussion, and it must be included. Answer questions such as: how could these findings generate a contribution to road designs? Which are the implications of these results for road safety, for the economy, for global climate change, for a road user to choose one car instead of another?

Response 6: Thank you very much for your advice. On the one hand, this work predicts the fuel consumption of vehicles with a speed over 120km/h, and finds out the difference in fuel consumption performance of different vehicle types. The prediction results can provide a reference for road users to choose more fuel-efficient vehicles. By choosing the most fuel-efficient and fuel-efficient vehicle, consumers will spend less on fuel and save fuel. Automobile exhaust pollution is an important factor contributing to global climate change, and low fuel consumption will lead to a relative reduction in vehicle exhaust emissions, thus reducing the emission of harmful gases such as particulate matter, nitrogen oxide and carbon dioxide. In addition, the research results can predict the use cost of a superhighway with a design speed of more than 120km/h, providing a basis for demonstrating the feasibility of the superhighway. I have added this to the conclusion in the manuscript.

Round 2

Reviewer 1 Report

After reading the amendments and additions proposed by the authors, I state as follows:

  1. The authors have made little reference to the Reviewer's remark and comments.
  2. The article may be considered as a report on the conducted research, and not as a scientific article showing a research problem, its solution and recommendations.
  3. I have not received an answer to the remarks and comments made in the previous review, in particular:   a)  Remark No. 2b, 2d, 2e,                                         b) Remark No. 3c and 3d,                                               c) Remark No. 4c,                                                       d) Remark No. 5a, 5b and 5c,                                     e) Remark No. 6a.
  4. Without supplementing and developing the problems mentioned in the remarks, and in particular remarks no. 2d, 5a, 5b and 6a, the reviewed article should not be published.

Author Response

Response to Reviewer 1 Comments

After reading the amendments and additions proposed by the authors, I state as follows:

Point 1. The authors have made little reference to the Reviewer's remark and comments.

Response 1: I'm sorry for the problem caused by my negligence. After careful consideration of your suggestions, I have made serious modifications to the corresponding parts of the manuscript.

Point 2. The article may be considered as a report on the conducted research, and not as a scientific article showing a research problem, its solution and recommendations.

Response 2: Currently, Chinese law stipulates that the speed limit on highways should not exceed 120km/h. However, with the great improvement of road construction technology and vehicle performance, as well as the successful operation experience of foreign expressways, it is possible for China to build superhighways with a design speed of more than 120km/h. Moreover, in 2018, The Hangzhou-Shao-yung Expressway began to build, which is the first designed expressway with a speed of 150km/h in China. And it is expected to be completed and open to traffic in 2022.The completion of this project will accumulate valuable experience for China's construction of superhighways designed to exceed 120km/h. This paper studies the vehicle fuel consumption of different models with a speed of more than 120km/h, and establishes a fuel consumption prediction model, which can estimate the fuel consumption cost of vehicles running on the superhighway in the future and provide reference Suggestions for car buyers. In addition, the next step may be to further study the exhaust emissions of different models through fuel consumption.

Point 3. I have not received an answer to the remarks and comments made in the previous review, in particular:   

  1. a)  Remark No. 2b, 2d, 2e

2b. description of the research methods used,

2d. description of the results and generalizations in the form of selected models of fuel consumption (FC) from the set of the most important factors and models of passenger car (PE) efficiency, in particular models for vehicles traveling at high speed (close to 200 km / h).

2e. The methods used to forecast the fuel consumption of passenger cars.

  1. b) Remark No. 3c and 3d, 

3c. Methods of obtaining data on the fuel consumption of individual vehicles and analysis of the accuracy of measuring fuel consumption,

3d. Distributions of data sets in individual speed classes together with their parameters.

  1. c) Remark No. 4c, 

  4c. For curvilinear functions, additional measures of function matching should be used (e.g. mean square error),

  1. d) Remark No. 5a, 5b and 5c,

   5a. Adoption of the fuel consumption model developed on the basis of a limited number of data (9 data) and a limited speed range (40 - 120 km) for extrapolation of data to fuel consumption models for vehicles developing speeds up to 180 km / h

5b. Lack of analysis of fuel consumption prediction errors when using of the cubic curves models, these models can cause quite large errors at high speeds.

5c. Lack of verification and validation of proposed predictive models. Proposed predictive models can be treated as preliminary to proper research.

  1. e) Remark No. 6a.

6a. Discussion of the results obtained and models of fuel consumption by motor vehicles, in particular against the background of researchers from other countries.

Response 3:

a).

2b. I have explained the description of domestic and foreign automobile fuel consumption and modeling methods in section 2.2 of the manuscript.

2d. The performance of the engine has the greatest impact on the fuel consumption of the vehicle at high speed. Generally speaking, the larger the displacement of the engine, the better the performance. The greater the output power, the better the dynamic performance of the vehicle. However, if the engine displacement is large, it may consume more fuel, and the fuel consumption of the car at high speed will also increase. Especially for vehicles with driving speed close to 200km / h, such as Ruicheng CC and 2019 Honda Civic. In 2018, Ruicheng CC realized the release test at the speed of 200 km / h in the high ring runway. The car is equipped with ACC adaptive cruise, PAB active brake and LDW lane departure warning function, which makes the overall automatic driving level of the car to L2 level, which can ensure the safety of the vehicle under the premise of providing comfortable driving. In addition, the 1.5T turbocharged engine equipped with Ruicheng CC has a maximum power of 156 HP (115 kW) / 5500rpm and a maximum torque of 225 nm / 2000-4000rpm, which makes the car have enough power to drive at high speed. From the perspective of driverless technology and engine performance, the driving safety of vehicles with a speed of more than 120km / h can be guaranteed, and there is enough power for them to drive at high speed. Therefore, it is meaningful to study the fuel consumption of vehicles over 120km / h. In addition, with the vigorous promotion of new energy vehicles, such as pure electric vehicles and hybrid electric vehicles, they have the common characteristics of clean, efficient, energy-saving and environmental protection. The popularization of new energy vehicles can offset the excessive energy consumption caused by ultra-high speed driving, and can effectively solve the energy consumption problem caused by the development of superhighway. I have explained this part in Section 2.2 of the manuscript.

2e. I have explained in lines 224~230 of the manuscript. There are also many methods used to predict passenger vehicle fuel consumption, such as artificial neural network method, multiple linear regression, least square method and simulation modeling method, etc. However, different methods have different advantages and disadvantages, which must be selected according to different vehicle types, different road conditions, and different driving modes. Most importantly, the prediction results should be verified and error analyzed, and necessary corrections should be made to ensure the accuracy of the prediction results.

b).

3c. First of all, we use the fuel consumption meter with the measurement accuracy of 0.01ml for the collection of fuel consumption test data. During the test, we connect the flow meter of the fuel consumption meter to the fuel supply pipeline of the vehicle engine under test. All the gasoline that has been burned to the engine will be measured through our fuel consumption meter. Then the fuel flow measured by the fuel consumption meter is the actual fuel consumption of the engine. The Test of Easy Car Platform adopted by the connection method to test the fuel consumption meter, can fundamentally solve the problems of fuel consumption measurement is allowed to jump shot, and considering the physical properties of objects "heat bilges cold shrink", we adopt the national standard correction formula is given for the final data for correction of fuel consumption test, objective and accurate fuel consumption test results are obtained. By these means, we will minimize the error of fuel consumption test results. This part has been mentioned in section 3.1 of the manuscript.

3d. The fuel consumption data measured in this study is directly calculated as the average value. That is, each car travels 500m at a constant speed, and it is tested back and forth for 4 times. The arithmetic average of the fuel consumption data measured for the four times is taken as the final constant speed fuel consumption of this speed level. In this way, the random error can be reduced and the accuracy of test data can be guaranteed. This part is explained in lines 247-253 of Section 3.1 of the manuscript.

c).

4c.Yes. In Section 5.1, I calculated the respective mean square error of each prediction model and found that the mean square error of each model was less than 0.1.

d).

5a. Because of limited fuel consumption data acquisition, only when the speed of 40 ~ 120 km/h of the vehicle fuel consumption, may have some error about the result of the forecast. but I establish the mean square error (mse) analysis for each prediction model of fuel consumption model, discovered that all the fuel consumption prediction model of mean square error is less than 0.1, the mean square error (mse) of the different fuel consumption prediction model are shown in table 15.It indicates that the established fuel consumption prediction model has a small deviation between the predicted fuel consumption value and the actual value, so the fuel consumption can be predicted under high-speed conditions. I have explained this part in Section 5.1 of the manuscript.

5b. Same as above 5a, since the mean square error is less than 0.1, indicating that fuel consumption prediction is relatively accurate.

5c. For the proposed fuel consumption prediction model, I conducted mean square error analysis in Section 5.1 and found that the mean square error of each prediction model was less than 0.1, indicating that the established fuel consumption prediction model was more accurate and the deviation between the predicted fuel consumption value and the actual value was small. In addition, in 2018, construction of China's first Hangzhou-Shao-yung expressway with a speed limit of 150km/h has begun and will be completed and open to traffic in 2020.When the highway is built, if conditions permit, it can be carried out on the actual vehicle road experiment and verified with the established fuel consumption prediction model.

e).

6a. Scholars from some European and American countries have also studied automobile energy consumption in many aspects, such as the relationship between engine performance, driving strategy, engine control and vehicle fuel consumption. The results show that the nonlinear model is most suitable for the changing law of vehicle fuel consumption. In addition, when the speed exceeds 120km/h, the fuel consumption of the vehicle increases with the increase of the speed. The results of European and American studies are similar to the cubic curve model established in this paper and the obtained relationship between speed and fuel consumption. I have added this content to lines 532~540 of the manuscript.

Point 4.Without supplementing and developing the problems mentioned in the remarks, and in particular remarks no. 2d, 5a, 5b and 6a, the reviewed article should not be published.

2d. description of the results and generalizations in the form of selected models of fuel consumption (FC) from the set of the most important factors and models of passenger car (PE) efficiency, in particular models for vehicles traveling at high speed (close to 200 km / h).

5a. Adoption of the fuel consumption model developed on the basis of a limited number of data (9 data) and a limited speed range (40 - 120 km) for extrapolation of data to fuel sconsumption models for vehicles developing speeds up to 180 km / h

5b. Lack of analysis of fuel consumption prediction errors when using of the cubic curves models, these models can cause quite large errors at high speeds.

6a. Discussion of the results obtained and models of fuel consumption by motor vehicles, in particular against the background of researchers from other countries.

Response 4:

2d. The performance of the engine has the greatest impact on the fuel consumption of the vehicle at high speed. Generally speaking, the larger the displacement of the engine, the better the performance. The greater the output power, the better the dynamic performance of the vehicle. However, if the engine displacement is large, it may consume more fuel, and the fuel consumption of the car at high speed will also increase. Especially for vehicles with driving speed close to 200km / h, such as Ruicheng CC and 2019 Honda Civic. In 2018, Ruicheng CC realized the release test at the speed of 200 km / h in the high ring runway. The car is equipped with ACC adaptive cruise, PAB active brake and LDW lane departure warning function, which makes the overall automatic driving level of the car to L2 level, which can ensure the safety of the vehicle under the premise of providing comfortable driving. In addition, the 1.5T turbocharged engine equipped with Ruicheng CC has a maximum power of 156 HP (115 kW) / 5500rpm and a maximum torque of 225 nm / 2000-4000rpm, which makes the car have enough power to drive at high speed. From the perspective of driverless technology and engine performance, the driving safety of vehicles with a speed of more than 120km / h can be guaranteed, and there is enough power for them to drive at high speed. Therefore, it is meaningful to study the fuel consumption of vehicles over 120km / h. In addition, with the vigorous promotion of new energy vehicles, such as pure electric vehicles and hybrid electric vehicles, they have the common characteristics of clean, efficient, energy-saving and environmental protection. The popularization of new energy vehicles can offset the excessive energy consumption caused by ultra-high speed driving, and can effectively solve the energy consumption problem caused by the development of superhighway. I have explained this part in Section 2.2 of the manuscript.

5a. Because of limited fuel consumption data acquisition, only when the speed of 40 ~ 120 km/h of the vehicle fuel consumption, may have some error about the result of the forecast. but I establish the mean square error (mse) analysis for each prediction model of fuel consumption model, discovered that all the fuel consumption prediction model of mean square error is less than 0.1, the mean square error (mse) of the different fuel consumption prediction model are shown in table 15.It indicates that the established fuel consumption prediction model has a small deviation between the predicted fuel consumption value and the actual value, so the fuel consumption can be predicted under high-speed conditions. I have explained this part in Section 5.1 of the manuscript.

5b. Same as above 5a, since the mean square error is less than 0.1, indicating that fuel consumption prediction is relatively accurate.

6a. Scholars from some European and American countries have also studied automobile energy consumption in many aspects, such as the relationship between engine performance, driving strategy, engine control and vehicle fuel consumption. The results show that the nonlinear model is most suitable for the changing law of vehicle fuel consumption. In addition, when the speed exceeds 120km/h, the fuel consumption of the vehicle increases with the increase of the speed. The results of European and American studies are similar to the cubic curve model established in this paper and the obtained relationship between speed and fuel consumption. I have added this content to lines 532~540 of the manuscript.

Reviewer 3 Report

On the one hand, I see that the authors did indeed check the section of the text that I pointed out in my previous review as an example of repetitive style in the manuscript, and I appreciate that they acknowledged the issue and changed this part. On the other hand, I think the authors could still substantially improve the writing of their article. When I said that the text had many repetitions, I referred to the text as a whole, and even though the specific section I used as an example was fixed, there are many parts of the manuscript that still feel very heavy to read. For instance, check this (lines 62.69): “On the one hand, the established fuel consumption prediction model can provide reference for the fuel consumption prediction of vehicles […]In addition, by establishing fuel consumption prediction models of different vehicle types and comparing the differences of fuel consumption performance of different vehicle types, […] Finally, the difference in fuel consumption performance is also related to vehicle exhaust emissions, which can be used as a reference for studying exhaust emissions of different vehicle models.” The words “fuel consumption” or “fuel consumption prediction” are repeated so many times, and even in the last lines “exhaust emissions” is repeated twice already. Obviously these terms cannot be changed and substituted by synonyms, but there are different ways that the authors could use to make the reading easier. For instance: “the established fuel consumption prediction model can provide reference for predicting the amount of fuel vehicles will consume”, or “the prediction model for fuel consumption can provide references for predicting the consumption of fuel of vehicles”, etc. (In general, I also think that this sentence is a little bit repetitive in concepts as well… what else could a fuel consumption prediction model predict, if not the consumption of fuel? However, this is my personal opinion). Another way of avoiding repetition: “Finally, the difference in fuel consumption performance is also related to vehicle exhaust emissions, which can be used as a reference for studying such emissions in different vehicle models”. Please note that these are just examples, but the issue is the same throughout the whole text! I will not provide more examples here, because it is already taking up much space, but I think the authors are getting a general idea. Again, I acknowledge the efforts that the authors are making, and I think they absolutely have the ability to improve their manuscript (that is why I am pointing this out).

Now, moving on to the contents:

Please include references for lines 111 and 125 (For instance Redelmeier, D. A., & Bhatti, J. A. (2017). Princess Diana and Reduced Traffic Deaths in France and the United States. American journal of public health, 107(8), 1246–1248. https://doi.org/10.2105/AJPH.2017.303880”, or “Driscoll, R., Page, Y., Lassarre, S., & Ehrlich, J. (2007). LAVIA--an evaluation of the potential safety benefits of the French intelligent speed adaptation project. Annual proceedings. Association for the Advancement of Automotive Medicine, 51, 485–505”). The same applies to lines 189-206.

It is not necessary to establish the used software in the conclusions, since you already said that earlier in the text.

Apart from this, I think the text has improved for what concerns my previous comments, although I really miss a discussion section.

Author Response

Dear Editor,

Thank you very much for giving me another chance to revise my paper.

We hope that our responses can meet the requirements of your journal. If there is any problem, please give us another chance. We will try our best to modify it until we meet your requirements.

Yours sincerely,

Yong-ming HE
